# Sphingolipid/Ceramide Pathways and Autophagy in the Onset and Progression of Melanoma: Novel Therapeutic Targets and Opportunities

**DOI:** 10.3390/ijms20143436

**Published:** 2019-07-12

**Authors:** Michele Lai, Veronica La Rocca, Rachele Amato, Giulia Freer, Mauro Pistello

**Affiliations:** 1Retrovirus Center and Virology Section, Department of Translational Research and New Technologies in Medicine and Surgery, University of Pisa, 56127 Pisa, Italy; 2Virology Unit, Pisa University Hospital, 56127 Pisa, Italy

**Keywords:** melanoma, sphingolipids, ceramides, acid ceramidase, multidrug resistance, autophagy

## Abstract

Melanoma is a malignant tumor deriving from neoplastic transformation of melanocytes. The incidence of melanoma has increased dramatically over the last 50 years. It accounts for most cases of skin cancer deaths. Early diagnosis leads to remission in 90% of cases of melanoma; conversely, for melanoma at more advanced stages, prognosis becomes more unfavorable also because dvanced melanoma is often resistant to pharmacological and radiological therapies due to genetic plasticity, presence of cancer stem cells that regenerate the tumor, and efficient elimination of drugs. This review illustrates the role of autophagy in tumor progression and resistance to therapy, focusing on molecular targets for future drugs.

## 1. Introduction

Melanoma is a malignant tumor deriving from neoplastic transformation of melanocytes, occuring at all anatomic sites where melanocytes can be found. Cutaneous melanoma generates from skin melanocytes and is the most common type of skin cancer, but it may also develop on mucous membranes and in the eye at a lower frequency [1,2]. The incidence of melanoma has increased dramatically over the last 50 years [3]. Although it accounts for 5% of all skin cancers, cutaneous melanoma is responsible for 75% of skin cancer deaths. Early diagnosis leads to remission in 90% of cases of melanoma because the tumor is located superficially and allows complete surgical removal. Indeed, surgical excision alone allows positive prognosis of over 90% of subjects with localized melanoma. Conversely, for melanoma at more advanced stages, prognosis becomes more unfavorable; the five-year survival rate from diagnosis varies between 60% and 24% according to the American Cancer Society. Advanced melanoma is often resistant to pharmacological and radiological therapies due to genetic plasticity, presence of cancer stem cells that regenerate the tumor, and efficient elimination of drugs. In this review, we will discuss the role of autophagy in tumor progression and resistance to therapy and examine a few metabolic pathways as potential targets to develop adjuvants or novel drugs overcoming resistance to therapy.

## 2. Molecular Signatures of and Current Therapies against Melanoma

Because of the intrinsic and remarkable ability of melanoma to develop resistance to therapy, intense efforts have been put to define novel approaches to treatment and pinpoint molecular pathways and signatures correlated to its progression. In this line of thought, the development of B-RAF inhibitors stemmed from the observation that the proto-oncogene B-Raf, encoding a serine/threonine protein kinase involved in cell signaling and directing cell growth [4], is mutated in a number of human cancers [5]. In addition, B-RAF mutations are present in benign nevi as well as in dysplastic ones, where the most common mutation is V600E [6,7]. B-RAF inhibitors showed adjuvant efficacy in combined treatment with standard chemotherapy in patients with melanoma; disappointingly, however, after less than six months, disease relapse was resistant to inhibitors, and consequently to therapy.

B-Raf is part of the mitogen-activated protein kinase (MAPK) cascade, which regulates complex cellular programs like proliferation, differentiation, development, transformation, and apoptosis [8,9]. It is therefore understandable why B-Raf plays a crucial role in the early stages not only of melanoma, but also in benign melanocytic neoplasms [10].

Recently, several lines of evidence indicate a pivotal role for the transcription factor associated with microphthalmia (MITF) in the onset of melanoma. MITF acts in the MAPK pathway, it regulates the development and differentiation of melanocytes, and it allows melanoblast survival [11,12]. Upregulated levels of MITF are associated with cell cycle arrest and differentiation, whereas downregulated levels lead to cell cycle dysregulation and apoptosis. Regulation of cell cycle by MITF is thought to occur primarily through interaction with p16 and p21 proteins, two onco-suppressors inhibiting cyclin-dependent kinases (CDK), and thereby triggering cell cycle arrest and apoptosis. Both onco-suppressors are frequently found mutated or deleted in primary tumors [13]. MITF, however, has multiple other roles, as it interacts with CDK2 and Bcl-2 proteins, promoting cell survival [11]; it also regulates transcription of ASAH1. As described in further detail in the following sections, ASAH1 encodes the enzyme acid ceramidase (AC) that controls sphingolipid metabolism and modulates the phenotypic switch of melanoma cells [14,15]. In melanoma, low levels of MITF are accompanied by low levels of ASAH1 expression, which correlates with invasive behavior and worse prognosis [11,12,15]. Furthermore, MITF knockout deranges the cell cycle and leads to cell death [15].

Another major player in the development of melanoma is phosphatidylinositol (PI)-3-kinase (PI3K) that regulates cell growth and survival, in combination with the factors mentioned above and others. Its pathway can be activated by Ras or by the loss of phosphatase tensin homolog (PTEN), encoded by the homonymous tumor suppressor gene, which is commonly expressed in melanoma cells. PTEN represses the PI3K pathway by dephosphorylating PI molecules. Growth signals increase intracellular levels of PI3 with a consequent phosphorylation and activation of protein kinase B (AKT) [16]. The increase in AKT expression found in melanoma is also observed in 50% of dysplastic nevi and 70% of metastatic melanomas [16]. In some 30–50% of melanomas, a loss of heterozygosity is described in the region of chromosome 10, where PTEN gene maps [17]. Inactivation of PTEN by deletion or mutation leads to constitutive activation of this pathway [16,17].

It is known that mutations of PTEN are rarely found in early stage melanomas [18,19], suggesting that activation of PI3K pathway is responsible for the development of late-stage melanoma, invasion, and metastasis [20]. Several studies show that mice with melanocytes presenting the above-mentioned V600E B-Raf mutation develop benign melanocytic hyperplasia that does not progress to melanoma until a subsequent mutation of PTEN transforms hyperplasia to fully malignant and metastatic melanomas [21]. This is probably why mutations of PTEN/ATK pathway occur later during tumor progression.

Melanoma therapy is mostly immunological, consisting of the administration of interleukin-2 (IL-2), unfortunately associated with toxicity and low response, or interferon-α (IFN-α) which showed some benefits. However, among the chemotherapeutic agents approved for advanced melanoma, dacarbazine, carmustine, Taxol, and cisplatin showed some efficacy in the treatment of metastatic melanoma. Other theoretical or current therapeutic approaches focus on genes involved in cell proliferation that are found mutated in melanoma. Among them, N-RAS, the first oncogene identified in melanoma, and B-RAF are also considered important therapeutic targets because it has been shown that one-third of all human cancers have oncogenic mutations in the small GTPase RAS family, which may therefore play a pivotal role in tumorigenesis [22]. Among the first inhibitors developed, a RAS farnesyl-transferase inhibitor blocks the post-translational modification of RAS. This inhibitor is not effective as a single treatment but works well in combination with Cis-Platinum. The B-RAF inhibitor, vemurafenib, has been shown to be effective in the treatment of patients with advanced melanoma and V600E mutation. Unfortunately, this drug has a low genetic barrier to resistance, as this treatment becomes ineffective very soon after the first months of therapy in many patients [23]. The bottom line of melanoma therapy is not, therefore, the restricted number of therapeutic options, but rather the extraordinary ability of the tumor to acquire resistance to single and combined chemotherapy, a feature called multidrug resistance (MDR).

## 3. Molecular Basis of Multidrug Resistance

MDR in melanoma is a major obstacle for its successful treatment by chemotherapy. Inherent or acquired resistance of melanoma cells to cytotoxic compounds has been the subject of extensive research, which has elucidated different molecular mechanisms. The best understood examples in this respect is overexpression of the ATP-binding cassette (ABC) transport proteins, such as P-glycoproteins (P-gp), and the multidrug resistance-related proteins (MRPs) [24]. ABC proteins are capable of conveying a wide variety of substrates across cellular membranes, including chemotherapeutic drugs. Several studies have reported that inhibiting these transporters might reverse MDR [24,25,26]. This task is, however, difficult to achieve in melanoma cells because they express a wide variety of ABC proteins, including ABCA9, ABCB1/5/8, ABCC2, and ABCD1, frequently found implicated, either alone or in combination, in the MDR phenotype [27,28].

Despite the diversity in chemical structure and molecular targets, many (if not all) cytotoxic drugs eventually induce apoptosis. In principle, therefore, any cellular mechanism that hampers the apoptotic signal cascade may contribute to the strong resistance to chemotherapeutic agents exhibited by melanoma cells. Indeed, it is well established that low levels and activity of key apoptotic molecules, such as p53 and members of the Bcl-2 family, can determine chemoresistance [29,30], and novel therapeutic approaches are indeed focusing on restoring the activation of p53-dependent apoptosis [31,32].

## 4. Sphingolipids and Ceramides

Ceramides are a class of bioactive lipids that regulate senescence, apoptosis, and autophagy [33]. They comprise more than 200 chemically and functionally distinct molecules that can be produced through de novo synthesis or by cleavage of preformed sphingolipids [34,35,36,37]. The de novo pathway occurs on the cytosolic side of the endoplasmic reticulum by condensation of a serine and palmitoyl coenzyme-A by the enzyme serine palmitoyltransferase (SPT). This eventually leads to the formation of ceramides, after a series of reactions. The degradation pathway is carried out by sphingomyelinase (SMase) starting from sphingomyelin, whereas, in the recycle pathway, sphingosine is acylated by the enzyme ceramide synthase with the formation of C2-ceramide. Once ceramides are formed, various enzymes act to maintain their intracellular levels.

Ceramides are the most abundant cellular sphingolipids and important enhancers of the apoptotic program [14,33]. As shown in Figure 1, ceramides create a pro-apoptotic environment because they regulate cell cycle arrest, apoptosis, and senescence; these molecules are also implicated in inflammation, differentiation, and autophagic regulation. Administration of chemotherapeutic agents, such as daunorubicin, camptothecin, and etoposide, leads to the formation of ceramides [38,39,40]. Therefore, cellular control of ceramide levels may be a critical factor influencing drug resistance.

The conversion of ceramides into more complex molecules (e.g., glucosylceramides (GlcCer)) allows them to decrease their intracellular levels and, as a consequence, to block their pro-apoptotic modulation. Based on the observations that ceramides trigger apoptosis and that the lipid metabolism in most tumors is deranged [41], ceramides are considered tumor-suppressing molecules. In keeping with this idea, many MDR cell lines exhibit increased levels of GlcCer when compared to their drug-sensitive counterparts [42,43]. Since the in vitro finding has been confirmed ex vivo (e.g., in tumor biopsies), the level of GlcCer has been proposed as a clinical marker for MDR [44]. It should be noted, however, that lack of GlcCer synthase, which physiologically gauges the GlcCer level, does not sensitize melanoma cells for chemotherapy, suggesting that the accumulation of GlcCer occurs through other pathways as well [45].

Contrary to ceramides, sphingosine-1-phosphate (S1P) contributes to create an anti-apoptotic phenotype, favoring cell growth, cell motility, migration, and angiogenesis [46]. S1P, once secreted, acts as an autocrine/paracrine signal through the S1P receptors [47]. The activation of S1P receptor mediates cell survival, migration, and angiogenesis [48].

Ceramides and S1P have attracted considerable interest in cancer therapy over the years because they influence the response to chemotherapy and radiotherapy [49]. Several tumors enhance the metabolism of ceramides by increasing the production of enzymes such as GlcCer synthase, sphingomyelin synthase, ceramide kinase, and AC that convert ceramides into molecules with a lower pro-apoptotic impact [14,15,49,50]. Among these enzymes, ceramidases are frequently found overexpressed as they convert ceramides into sphingosine, thus indirectly increasing S1P levels and transforming a pro-apoptotic signal into a cell survival one [15].

## 5. Sphingolipid Metabolism and Cancer

Sphingosine and ceramide generation is induced by cellular stress through the activation of de novo synthesis pathways, the hydrolysis of sphingomyelin, and by the salvage pathway to mediate pro-apoptotic stimuli [51]. Many tumors, including melanoma, increase ceramide metabolism mainly by the activity of GlcCer synthase, sphingomyelin synthase (SMS), ceramide kinase (CERK), and AC. Overall, these enzymes generate sphingolipids with pro-survival functions [52]. Ceramides are divided based on the number of carbons that compose the fatty acyl chain, ranging from 14 to 26. The de novo synthesis of ceramide by (dihydro)ceramide synthases generates ceramides with different fatty acyl chains that have different biological roles in cancer cells, for mechanisms that are still unclear. It is possible that these different biological activities depend on the differential subcellular localization in normal/cancerous cells. The conversion of sphingosine to ceramide is catalyzed by the salvage pathway. Interestingly, these reactions involve the same enzymes of the de novo ceramide synthesis. The ceramidase activity, together with the sphingosine kinase 1 and 2, converts ceramides to sphingosine-1-phosphate. This bioactive lipid is further hydrolyzed by S1P lyase to exit the sphingolipid metabolic circle. Ceramides are also a substrate to generate complex sphingolipids such as gangliosides and glycosphingolipids [53]. In these reactions, ceramides are first converted into glucosylceramides, which are the precursors needed for the synthesis of complex sphingolipids. Interestingly, patients with Gaucher disease, which is an autosomal recessive genetic disorder caused by mutations of the lysosomal glucocerebrosidase enzyme, have a higher risk (37-fold) to develop myelomas compared to the general population [54]. Moreover, it seems that these patients have increased chances of developing multiple consecutive cancers in their lifespan. It is still unclear if defects in the glucocerebroside metabolism may be involved in the development of melanoma [55].

## 6. Autophagy

Since apoptosis contrasts MDR, drugs triggering programmed cell death in MDR cancer cells would be valuable. Unfortunately, the variegated pathways and triggers tweaked by melanoma cells to inactivate apoptosis add considerable complexity to the development of such drugs.

A potentially exploitable mechanism to interfere with MDR is autophagy. Autophagy is a finely regulated mechanism that degrades damaged cellular material and converts catabolites into nutrients and precursor compounds for the cell. Through this process, cells ensure basal turnover of cellular organelles and constant supply of energy and macromolecular precursors [56]. Active autophagy is also named autophagic flux as it occurs as a continuum; this process can be distinguished in several phases, such as formation of the phagophore and the autophagosome, and lysosomal fusion. Autophagy has been described in several articles that thoroughly discuss the roles of many proteins and lipids [57,58,59], which are actively involved in each of the stages. Here, we will briefly discuss the core pathway.

Autophagy is induced in response to various stress stimuli, such as starvation, reactive oxygen species, hypoxia, infection, and drugs. What causes the initial step that leads to the formation of the phagophore is still mostly unclear.

### 6.1. Phagophore Formation

This is an initial event following autophagy induction. Phagophore is a small cup-shaped membrane whose origin has been a matter of debate for years. Recent evidence indicates that the lipid membranes forming the autophagosomes are generated through a mechanism directed by autophagy-related (ATG) proteins that are located in the cytoplasm. Over 15 ATG proteins take part in autophagophore formation. Another important player is the PI3K enzyme that produces the phosphatidylinositol 3-phospate (PI3P) and regulates the trafficking of proteins and vesicles (Figure 2) [60]. Conversely, the lipid species and membrane modelling involved in phagophore composition, shape, and size are still unclear [61]. In higher eukaryotes, phagophore nucleation occurs on a preexisting membrane [62,63,64].

### 6.2. Autophagosome Formation

The mechanism involved in tethering and fusion of incoming membranes to phagophore seems to be regulated by PI3K and the Unc-51-like autophagy activating kinase (ULK). These enzymes contribute to the formation and stabilization of phagophore curvature, through the production of PI3P [65]. In this phase, ATG and microtubule-associated proteins 1A/1B light chain 3B (LC3) regulate the double membrane generation. LC3 is a central protein in autophagy and crucial for autophagosome biogenesis, hence LC3 is also used as a marker of autophagosome formation. During autophagosome formation, LC3 is hydrolyzed by ATG proteins and binds phosphatidylethanolamine [66]. This conjugated compound called LC3-II is believed to be involved in the expansion of autophagosome membrane and to be necessary for the fusion events taking place thereafter [67,68].

### 6.3. Lysosome Fusion

After complete fusion of the expanding ends of the phagophore and generation of the autophagosome, the next step consists of the fusion of the autophagosome with the lysosome to form the “autolysosome” (Figure 2) [69]. Fusion of the two organelles is mediated by the small G protein Rab7 and cellular cytoskeleton; indeed, the administration of nocodazole, which destroys the microtubule structure, blocks autophagosome-lysosome fusion [70]. Cloroquine, an FDA-approved antimalaric drug, blocks autophagic flux by preventing lysosome fusion with the autophagosome [71] and is currently being tested in clinical trials aimed at sensitizing tumors to chemotherapeutics through autophagy inhibition.

To some extent, autophagy can therefore be exploited to enhance treatment efficacy of MDR cancer cells. This is a narrow path, because we need to keep in mind that autophagy can also contribute to MDR development by accelerating drug degradation during chemotherapy. It is unclear whether autophagy also influences ATP-dependent drug efflux by ABC transporter proteins. Recent studies have shown that expression levels of ABCB1, considered part of the MRP network, and autophagic markers LC3, Beclin1, and Rictor are positively correlated, suggesting a possible connection between autophagy and the ABC transporter system [72]. In keeping with this hypothesis, tumor patients with poor prognosis exhibited a concomitant increase in levels of autophagy and related markers and ATP-dependent drug efflux, possibly synergizing to the development of a robust MDR phenotype [73,74,75].

Autophagy is relevant in cancer and, in this context, it is thought to play a dual role. On the one hand, it has been shown to prevent cancer development; on the other, once cancer is established, an increase in autophagic flux enables tumor cells to thrive, expand, and resist chemotherapy [45,56]. Indeed, autophagy can be also considered a mechanism that protects tumor cells from chemotherapy and facilitates MDR development [76]. Inhibition of autophagy hampers the active elimination of drugs and can, in principle, re-sensitize resistant cancer cells and enhance the efficacy of chemotherapeutic agents. Thus, depending on metabolic or therapeutic stimuli, the modulation of autophagy has either pro-death or pro-survival effects.

The relationship between tumor cell survival and autophagy can only, in part, be explained by the role of autophagy in protecting cells from apoptosis [56,77]. Autophagy does not always spare tumor cells from apoptosis nor is it beneficial for cells at all times. For example, excessive autophagy, a phenomenon characterized by the disproportionate elimination of organelles, leads cancer cells to “autophagic cell death” [78]. Furthermore, autophagy can promote but also inhibit apoptosis in the same tumor cell population in response to various stimuli, such as the death receptor agonists CD95 ligand (CD95L) and tumor necrosis factor-related apoptosis-inducing ligand (TRAIL) [79]. The latter autophagic pathways can be envisioned as targets of antimelanoma therapy for future drugs.

## 7. Ceramides, Sphingolipids, and Autophagy

As mentioned, ceramides are at the crossroads between protective and lethal autophagy (Figure 3). In protective autophagy, ceramides may inhibit AKT protein kinase by activating pyrophosphatase protein 2A (PP2A); AKT interferes with the mammalian target of rapamycin (mTOR) kinase pathway, the main negative regulator of autophagy [80]. In other circumstances, ceramides can induce stress to the endoplasmic reticulum (ER), resulting in activation of the ER stress sensor inositol-requiring element 1 (IRE-1). Activation of this stress pathway leads to apoptosis within a certain time span [81]. Ceramides can also induce lethal autophagy, an irreversible process called type II cell death. In this case, they induce the activation of phosphatase-associated proteins (CAPP) that inhibit mTOR [82]. Finally, ceramides are also involved in the increase of autophagic protein Beclin-1 through activation of the transcription factor c-Jun. It is still unclear how ceramides might play this dual regulation of autophagic flux, but the induction of lethal autophagy may depend on the existence of cellular microdomains where ceramides localize to activate different biological roles.

Ceramides are hydrolyzed to sphingosine by ceramidases. Sphingosine is then converted to S1P by sphingosine kinase-1 (SPHK1) or sphingosine kinase-2 (SPHK2). S1P also regulates autophagy but, unlike ceramides, induces a protective autophagic process [83]. Given the opposite effects of ceramides and sphingosine on autophagy, AC, an enzyme playing a key role in the conversion of ceramides into sphingosine, is pivotal in cell activity and fate.

## 8. The Acid Ceramidase Enzyme

At least five different human ceramidases have been recognized. These are encoded by distinct genes, which are classified according to their optimal pH for enzymatic activity: AC (N-acylsphingosine deacylase), alkaline ceramidase, further divided into three isoforms, and neutral ceramidase. AC was identified by Gatt in 1963 and is encoded by ASAH1 that maps to chromosome 8 (p21.3-22). AC is initially translated as a precursor of 53-55 kDa that is then cleaved in the subunits α (13 kDA) and β (40 kDa). AC is a lysosomal hydrolase that catalyzes the hydrolysis of ceramide into sphingosine and free fatty acids [84]. Ceramides with an acyl chain of 6 to 16 carbons are the preferential AC substrates [85]. AC deficiency causes Farber disease, a rare lysosomal disease characterized by deformation of the joints, subcutaneous nodules, and progressive hoarseness. It is an autosomal recessive disorder further distinguished into seven subtypes depending on severity and tissues involved. Seventeen different mutations of ASAH1 have been identified in patients with Farber disease. Most of these are single nucleotide substitutions, but small deletions and missense mutations generating a truncated protein have also been found. Farber disease has no cure. In 2002, a AC knockout mouse model of the disease was created, but unexpectedly, the homozygous animals died at an embryonic stage, suggesting that AC also plays an important role during development, at least in mice [81]. In such a knockout model, developing embryos died of apoptosis, possibly due to increased levels of ceramides [86].

Several studies have shown that AC is overexpressed in many tumors [15,87,88,89,90]. Concerning melanoma, Realini and coworkers have shown that AC is expressed at higher levels in proliferative melanoma cell lines compared to skin cells, such as keratinocytes, fibroblasts, and melanocytes [33]. In addition, the expression of ASAH1 is greater in proliferative than invasive melanomas. In addition, the prostate tumor cell line DU154, which also overexpresses AC, shows greater proliferation even in the absence of nutrients and increased in vivo tumorigenicity and cell migration compared to other tumor cells exhibiting normal AC levels [91,92]. Resistance to starvation may be due to the activation of protective autophagy, suggesting that AC can influence the response to chemotherapeutics not only by affecting apoptosis. In a paper published in 2011, Turner and colleagues showed that also prostate cancer tumor lines overexpressing AC have high autophagic activity. These tumor lines show resistance to conditions of nutrient deprivation and exhibit high expression of the LC3-II protein. From these results, the authors concluded that autophagy contributes to creating an “insult-ready” phenotype and AC overexpression makes these cells more resistant to stress [93].

The observation that AC is overexpressed in pediatric brain tumor lines resistant to chemo- and radiotherapy [94] led to the development and testing of carmofur, a potent AC inhibitor and chemotherapeutic agent in these neoplastic cells. Following treatment, the tumor cells accumulated high levels of ceramides and died off [95,96,97,98,99].

Studies on glioblastoma cells have shown that gamma radiation increases AC activity in these cells [100]. In addition, treatment of these tumor cells with an inhibitor of AC, N-oleylethanolamine (NOE), increases ceramide production and apoptosis in response to gamma radiation, suggesting that AC exerts a protective role during radiotherapy by reducing ceramides. Treatment with daunorubicin (DNR) was also shown to induce overexpression of AC in hepatocellular lines [101]. To investigate this phenomenon, the authors knocked down AC or added exogenous S1P to these tumor cells and discovered that the ablation of AC increased sensitivity to DNR through the induction of apoptosis; in contrast, SP1 addition promoted tumor cell survival by the inhibition of apoptosis [102]. The mechanism through which AC overexpression favors tumor survival has not yet been clarified. It is possible that cells resist better to various stress stimuli in the presence of low levels of ceramides and/or high levels of S1P produced by the AC-catalyzed hydrolysis of ceramides.

We have previously shown that ceramide metabolism and AC control melanoma progression [14]. This finding rendered AC an important therapeutic target and, as a result, several pharmacological inhibitors and gene silencing systems have been developed to evaluate the importance of AC in tumor development. Over the years, in addition to NOE, other ceramide-like compounds have been developed to inhibit AC activity, for example, (1S, 2R) -D-erythro-2 (N-myristoylamino) -1- phenyl 1 - propanol (De-MAPP) and its B13 analog, which potently inhibit AC in vitro in human melanoma cell lines. Unfortunately, these inhibitors have modest efficacy in vivo.

Bedia and collaborators (2011) used dacarbazine, an imidazole carboxamide commonly used in the treatment of metastatic melanoma and with low efficacy against the invasive melanoma cell line A375 [59]. This study evidenced that sensitivity to the drug depended on the levels of AC expression. A few years ago, Realini and collaborators showed that carmofur, a chemotherapeutic agent used in Japan since 1981 against colorectal cancer, is a potent AC inhibitor in A375 melanoma cell line [103]. Treatment of this cell line with dacarbazine led to significant loss of AC activity, increment of C16 and C18 ceramidase, and concomitant restoration of sensitivity to currently used melanoma drugs [49]. Encouraged by these studies, Realini and collaborators developed the carmofur derivative ARN 14988 inhibiting AC activity in a time/dose-dependent manner in both invasive and proliferative melanoma cell lines. Compared to carmofur, ARN14988 lacks 5-Fluoro-uridine, has lower cytotoxic effects per se but enhances synergistically the cytotoxic effect of standard chemotherapeutics. In the proliferative melanoma line G361, for instance, ARN 14988 acted in a synergistic way with 5-Fluorouracil, Taxol and vemurafenib [104]. For these features, ARN 14988 is reputed more suitable as an adjuvant rather than a bona fide antineoplastic drug.

Derangement of the ceramide–sphingolipid pathway also alters cell cycle. Only 4% of the above-mentioned A375 cells knocked out of ASAH1 (A375 ASAH1-null cells) entered phase G2, compared to 25% of controls, and about 50% of ASAH1-null cells remained in phase G1 or in phase S (46%) [14]. For this reason, the interaction of ceramides with the cell cycle cascade is thought to occur through CDK inhibitor p27, which controls the transition from G1-S to G2-mitosis phase [105]. AC ablation altered not only the levels but also the type of ceramides, with an increase in long-chain molecules such as C14:0, C16:0, and C18:0. Furthermore, a higher percentage of ASAH1-null cells underwent apoptosis compared to wild-type cells [14]. Taken together, in vitro findings support the idea that AC overexpression favors resistance to chemotherapeutic treatment by decreasing the levels of ceramides, which, in turn, lead to cell death. As an additional effect, the imbalance of ceramide–sphingolipid levels, involved in the autophagic process and embedding pro-apoptotic (ceramides) or anti-apoptotic (sphingosine and S1P derivative) signals, helps cancer cells to withstand stress. This makes AC a potential target to develop either chemotherapeutic agents or adjuvant molecules that make tumor cells less prone to survive in the face of antitumor treatment.

## 9. Conclusions

Autophagy is coming out as a process that dictates opposite fates for cells: it can protect cells from apoptotic stimuli (protective autophagy) or drive cells to death (lethal autophagy), as illustrated in Figure 4. AC has been shown to participate in determining one outcome or the other as it regulates levels of molecules possessing pro- or anti-apoptotic effects. A future challenge might definitely be shedding light on what drives the cell into one or the other possible direction.

Tumors expressing high levels of AC and consequently lower ceramide levels show higher protection from stress stimuli, including chemotherapy. It is worth mentioning, however, that localization of AC in cancer cells is not limited to lysosomes, like in normal cells, but is also present in the cytoplasm [33]. Although the role and activity of cytoplasmic AC are still unclear, it is arguable that the adjuvant effect showed by some AC inhibitors depends on the different localization of the enzyme rather than its sole inhibition in tumor cells. An innovative approach to avoid and circumvent MDR is, therefore, the co-delivery of AC inhibitors and chemotherapeutic agents, aimed at disarming the complex apparatus at the disposal of cancer cells to blunt drug efficacy.

## Figures and Tables

**Figure 1 ijms-20-03436-f001:**
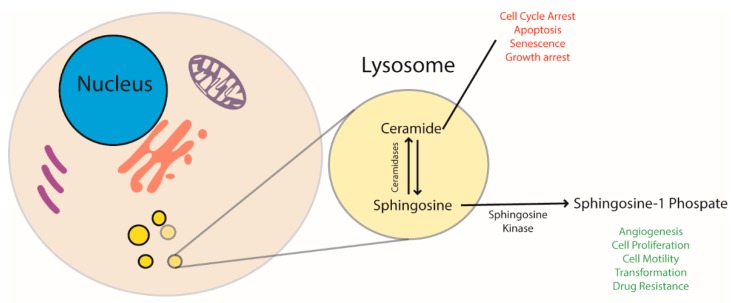
Ceramide and sphingosine-1-P regulate and stimulate opposite cellular pathways. The former has a pro-apoptotic effect and blocks cell replication, the latter stimulates cells to proliferate. AC balances ceramides and sphingosine levels. Since sphingosine is then converted to sphingosine-1-P by sphingosine kinase, AC has a direct role in regulating cell life.

**Figure 2 ijms-20-03436-f002:**
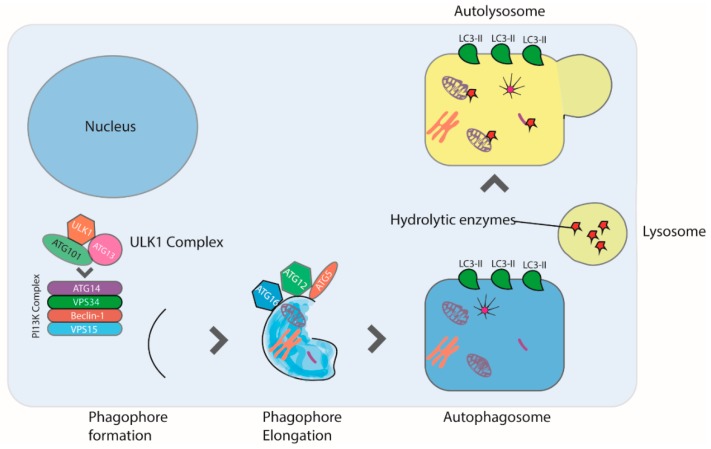
Autophagy is induced in response to starvation, reactive oxygen species, hypoxia, infection, and drugs. Autophagy begins with the formation of the phagophore, a step mediated by the ULK1 complex. Next, phagophore nucleation requires the PI3K complex, which consists of ATG14L, VPS34, Beclin-1, and VPS15 proteins. Autophagosome formation requires the lipidation of LC3 to form LC3II. Finally, the autophagosome fuses with a lysosome forming the autophagolysosome, leading to the degradation of the sequestered organelles and cytosolic proteins.

**Figure 3 ijms-20-03436-f003:**
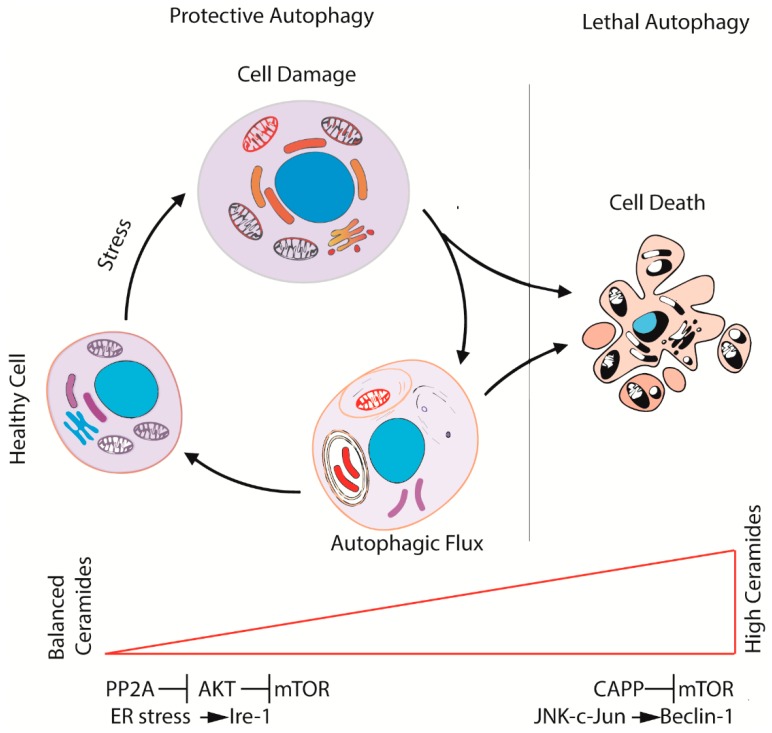
The ceramide–sphingolipid axis contributes to regulate the fate of cells during autophagy. Balanced levels of ceramides activate autophagy through the inhibition of AKT and, consequently, of mTOR (protective autophagy). While the levels of ceramides increase, the over-activation of autophagy leads the cell to autophagic cell death through JNK-c-Jun and Beclin1 signaling.

**Figure 4 ijms-20-03436-f004:**
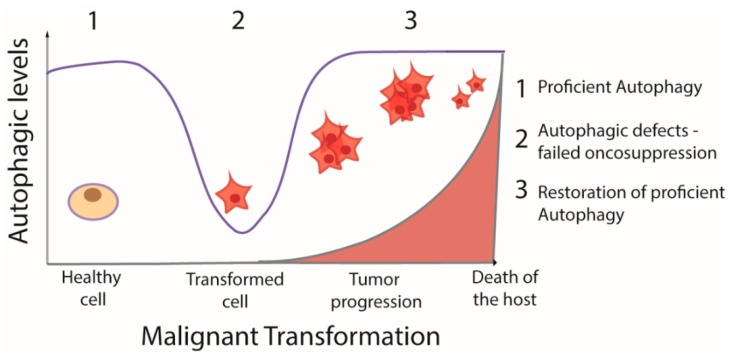
Schematic model of the role of autophagy during cancer progression. Proficient autophagy is blocked during cell transformation and restored in the advanced phases of tumor progression.

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
