# Peer review of "Sphingolipid/Ceramide Pathways and Autophagy in the Onset and Progression of Melanoma: Novel Therapeutic Targets and Opportunities"

_ijms, 2019, doi:10.3390/ijms20143436_

Round 1

Reviewer 1 Report

The present review article deals with some interesting summarization of the sphingolipid/ceramide components in autophagy and tumor cell behaviour of melanoma, providing prospective modulating targets.

Since they are consitituted on the cellular membrane in eukaryotes, the membrane function is important in cellulr modulation with microdomains and cellular receptors to mediate various phenotypes.

General emphasis has well been made in this eview. However, one major point should be added in the revision: the biosynthetic pathway of the sphinogolipids/ceramide is directly regulated to undergo their phenotype functions in cells such as melanoma.  Thus, they have to in brief add the biosynthetic points in cells. Please remind the melanoma is phenotypically expressing GD3 ganglioside on the surface, reserving the imporance of the sphingolipids on membrane.

This is an interesting and well-organized review when they revise it.

Author Response

… one major point should be added in the revision: the biosynthetic pathway of the sphinogolipids/ceramide is directly regulated to undergo their phenotype functions in cells such as melanoma. Thus, they have to in brief add the biosynthetic points in cells. Please remind the melanoma is phenotypically expressing GD3 ganglioside on the surface, reserving the imporance of the sphingolipids on membrane.

To address this point, we added paragraph 4 in the revised version.

Reviewer 2 Report

The paper by Lai et al. entitled "Sphingolipid/ceramide pathways and autophagy in the onset and progression of melanoma: novel therapeutic targets and opportunities" is a review paper on the current knowledge of melanoma focusing on phingolipid ceramide pathways. Overall, the paper is well structure and organized, with comprehensively reviewing literature. I think the paper will be of interest for the readers of the journal.

I would like to raise the following minor concerns.

1, In the abstract, the authors stated that the prognosis of advanced melanoma is worse with a 5-year survival rate of < 10%. This information should be updated. The prognosis of advanced melanoma has dramatically been improved.

2, It is difficult to read the letters in the Fig.2  because of the size and low resolution. Fig 2 should be improved.

Author Response

Point 1

In the abstract, the authors stated that the prognosis of advanced melanoma is worse with a 5-year survival rate of < 10%. This information should be updated. The prognosis of advanced melanoma has dramatically been improved.

The abstract has been revised according to the most recent data by the American Cancer Society.

Point 2

It is difficult to read the letters in the Fig. 2 because of the size and low resolution. Fig 2 should be improved.

Fig 2 has been completely changed and improved.